# PowerGraph: A power grid benchmark dataset for graph neural networks

## Abstract

Public Graph Neural Networks (GNN) benchmark datasets facilitate the use of GNN and enhance GNN applicability to diverse disciplines. The community currently lacks public datasets of electrical power grids for GNN applications. Indeed, GNNs have the potential to capture complex power grid phenomena over alternative machine learning techniques. Power grids are complex engineered networks that are naturally amenable to graph representations. Therefore, GNN have the potential for capturing the behavior of power grids over alternative machine learning techniques. To this aim, we develop a graph dataset for cascading failure events, which are the major cause of blackouts in electric power grids. Historical blackout datasets are scarce and incomplete. The assessment of vulnerability and the identification of critical components are usually conducted via computationally expensive offline simulations of cascading failures. Instead, we propose the use of machine learning models for the online detection of cascading failures leveraging the knowledge of the system state at the onset of the cascade. We develop Power-Graph, a graph dataset modeling cascading failures in power grids, designed for two purposes, namely, i) training GNN models for different graph-level tasks including multi-class classification, binary classification, and regression, and ii) explaining GNN models. The dataset generated via a physics-based cascading failure model ensures the generality of the operating and environmental conditions by spanning diverse failure scenarios. In addition, we foster the use of the dataset to benchmark GNN explainability methods by assigning ground-truth edge-level explanations. PowerGraph helps the development of better GNN models for graph-level tasks and explainability, critical in many domains ranging from chemistry to biology, where the systems and processes can be described as graphs. The dataset is available at `https://figshare.com/articles/dataset/PowerGraph/22820534` and the code at `https://anonymous.4open.science/r/PowerGraph/`.

## 1 Introduction

The lack of public Graph Neural Network (GNN) datasets for power grid applications has motivated the development of a new graph dataset. Power grid stability is crucial to modern society, and, therefore, power grids are designed to be robust under failures of different nature. Under particular conditions, however, the failure of critical components can trigger cascading outages. In the worst case, cascading failures spread into the full blackout of the power grid Andersson et al. (2005); Haes Alhelou et al. (2019). The complete understanding of complex events as cascading failures is therefore of uttermost importance. Such events are rare and historical data is scarce, therefore, we must rely on simulating cascading failures via computer models. The established traditional approach for cascading failure analysis is a quasi-steady state model, such as the OPA model Carreras et al. (2002), the Manchester model Nedic et al. (2006), and the Cascades model Gjorgiev et al. (2022). These models assess how the power grid responds after an outage is introduced in the grid. In fact, they simulate the complex behavior of the systemic responses and how a chain of successive failures (cascade) propagates in the grid. Since such tools are computationally intensive, they cannot be used by power grid operators for online detection of cascading failure nor for probabilistic risk analysis employing sequential Monte Carlo.

The shortage of historical blackout data and the high computational cost of current methods to simulate cascading failures in power grids highlight the need for machine learning models that

can detect cascading failures in almost real-time. Power grid operators, specifically transmission system operators (TSO), will greatly benefit from an online tool able to estimate the potential of cascading failures under given operating conditions of the power grid. The research community has presented new methods that employ machine learning algorithms for the online prediction of cascading failures. The proposed methods often do not generalize for diverse sets of failures Abedi et al. (2022); Aliyan et al. (2020). They are trained with datasets created with cascading failure models that often rely on the direct current (DC) power flow approximation Liu et al. (2020), less accurate than the alternate-current (AC) power flow. In addition to these limitations, the authors are not aware of publicly available datasets on the subject.

Within the realm of machine learning algorithms, GNN are convenient and powerful machine learning algorithms to model power grid phenomena, since graphs allow an intuitive representation of power grids. In Liao et al. (2021), the authors introduce how GNN have been employed for various applications in the field of power systems. Our paper focuses on fault scenario application, but we plan to extend it to power flow calculation in the future. On this topic, the authors of Yaniv et al. (2023) provide a review of GNN for power flow models in the distribution systems. The work in Varbella et al. (2023) shows that a GNN outperforms a feed-forward neural network in predicting cascading failures in power grids. To produce a large and complete dataset, we use Cascades Gjorgiev et al. (2022), an alternate-current (AC) physics-based cascading failure model. The model simulates the evolution of the triggering failures yielding the final demand not served (DNS) to the customers. We produce a power grid GNN dataset comprising a large set of diverse power grid states. The power grid state represents the pre-outage operating condition, which is linked to the initial triggering outage (one or more failed elements), referred to as the outage list. Each power grid state is represented as a graph, to which we assign a graph-level label according to the results of the physics-based model. The dataset is generated to suit different graph-level tasks, including multi-class classification, binary classification, and regression.

The presented graph property prediction dataset fills a gap according to the OGB taxonomy for graph dataset Hu et al. (2020a; 2021). Graph datasets are classified according to their task, domain, and scale. The task is at the node-, link-, or graph- level; the scale is small, medium, or large; and the domain is nature, society, or information. Our dataset comprises a collection of power grid datasets, which are designed for graph-level tasks, and their size ranges from small to medium Freitas et al.. Moreover, all the datasets in PowerGraph have the same number of features per node, and therefore, they can be utilized as one combined dataset to train GNN models. Table 1 reports the total number of graphs per power grid, the number of buses and branches in the grid, the number of loading conditions, and the number of outage lists simulated. The dataset fits the society domain, where no public GNN graph property prediction datasets are available Hu et al. (2020a), see Appendix A.1.

Table 1: Parameters of the AC physics-based cascading failure model for the selected four test power grids. A bus is defined as a node where a line or several lines are connected and may also include loads and generators in a power system. Transmission lines and transformers are defined as branches.

| Test system | # Bus | # Branch | # Loading conditions $n_{load\ cond}$ | # Outage lists $n_{outage\ lists}$ | # Graphs $N$ |
|---|---|---|---|---|---|
| IEEE24 | 24 | 38 | 300 | 43 | 12900 |
| UK | 29 | 99 | 300 | 132 | 39600 |
| IEEE39 | 39 | 46 | 300 | 55 | 16500 |
| IEEE118 | 118 | 186 | 300 | 250 | 75000 |

Other relevant GNN datasets for graph property prediction are the TU collection Morris et al. (2020) and the MoleculeNET Wu et al. (2018) dataset. Their application is natural science, particularly molecular graphs, i.e., molecules are represented as graphs to predict certain chemical properties. Publicly available power grid datasets such as the Electricity Grid Simulated (EGS) datasets Dua & Graff (2017), the PSML Zheng et al. (2021), and the Simbench dataset Meinecke et al. (2020) are not targeted to machine learning on graphs. In addition, both the EGS and PSML provide data for very small power grids, with 4 and 13 nodes respectively. Instead, Simbench focuses only on power system analysis in the German distribution and transmission grid, and the dataset is not designed for machine learning on graphs. In Nauck et al. (2022), the authors present new datasets of dynamic stability of synthetic power grids. They found that their GNN models, which primarily use emphasizes node regression, can predict highly non-linear targets from topological information. On the other

hand, PowerGraph, which uses graph-level tasks, does not address dynamic stability and relies on established real-world-based power grid models to predict the development of cascading failures. Overall, the dataset we provide fills a gap in the domain of GNN datasets for graph-level tasks Hu et al. (2020a) and is the only publicly available GNN dataset for power grids.

Besides benchmarking GNN models, the dataset is intended to be used for explainability methods. Therefore, we assign ground-truth edge explanations using the insights provided by the physics-based cascading failure model. As explanations, we consider the branches that have failed after the initial trigger, i.e., the cascading stage. In the field of explainability for GNN, there is to the best of our knowledge no existing real-world dataset with reliable ground-truth explanations Agarwal et al. (2023). There have been recent attempts to create a synthetic graph data generator producing a variety of benchmark datasets that mimic real-world data and are accompanied by ground-truth explanations Agarwal et al. (2023), as well as to provide atom-wise and bond-wise feature attribution for chemical datasets Hruska et al. (2022); Jiménez-Luna et al. (2022). However, none of these attempts provides real world data with empirical explanations. Here, we propose a real world dataset for GNN graph level tasks that has clear ground-truth explanations obtained from physic-based simulations.

This work provides a large-scale graph dataset to enable the prediction of cascading failures in electric power grids. The PowerGraph dataset comprises the IEEE24 Engineering (b), IEEE39 Engineering (c), IEEE118 Engineering (a) and UK transmission system NationalgridESO. These test power systems have been specifically selected due to their representation of real-world-based power grids, encompassing a diverse range of scales, topologies, and operational characteristics. Moreover, they offer comprehensive data with all the necessary information required for conducting cascading failure analysis. With PowerGraph, we make GNN more accessible for critical infrastructures such as power grids and facilitate the online detection of cascading failures. Our contributions are the following:

- We provide a data-driven method for the online detection of severe cascading failure events in power grids.
- We make the dataset public in a viable format (PyTorch Geometric), allowing the GNN community to test architectures for graph-level applications.
- The dataset includes several graph-level tasks: binary classification, multi-class classification, and regression.
- We provide explanatory edge masks, allowing the improvement of GNN explainability methods for graph-level applications.

The rest of the paper is organized as follows: Section 2 describes the physics-based model used to simulate cascading failure scenarios; Section 3 outlines the structure of the graph datasets; Section 4 reports the benchmark experiments of the different datasets; Section 5 describes the method used to benchmark explainability methods; and Section 6 concludes the article with a final discussion.

## 2 PHYSICS-BASED MODEL OF CASCADING FAILURES

We employ the established Cascades model Gjorgiev et al. (2022); Gjorgiev et al. for cascading failure simulations to produce the GNN datasets. Indeed, its application to the Western Electricity Coordinating Council (WECC) power grid demonstrates that Cascades can generate a distribution of blackouts that is consistent with the historical blackout data Li et al. (2018). Cascades is a steady-steady state model with the objective to simulate the power grid response under unplanned failures in the grid. For that purpose, the model simulates the power system's automatic and manual responses after such failures. Initially, all components are in service and there are no overloads in the grid. The system is in a steady-state operation with the demand supplied by the available generators, which produce power according to AC- optimal power flow (OPF) conditions Bouchekara (2014). The simulation begins with the introduction of single or multiple initial failures. Then, Cascades simulates the post-outage evolution of the power grid, i.e., identifies islands, performs frequency control, under-frequency load shedding, under-voltage load shedding, AC power flows, checks for overloads, and disconnects overloaded components. The model returns two main results: the demand not served (DNS) in MW and the number of branches tripped after the initial triggering failure. The simulation is performed for a set of power demands sampled from a yearly load curve. For each season of the year, an equal number of loading conditions are randomly sampled. We use a Monte-Carlo simulation to probabilistically generate outages of transmission branches (lines and transformers). We define the number of loading conditions and the size of the outage list. Therefore, we are able to simulate a large number of scenarios and thus create large datasets. Each scenario generated is a power grid state,

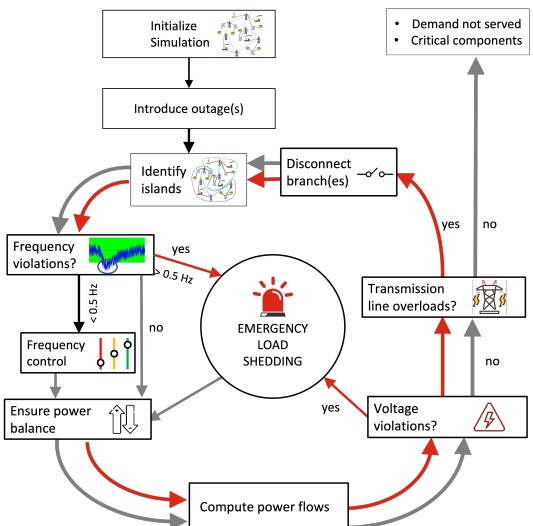

Figure 1: Workflow of the Cascades Gjorgiev & Sansavini (2022) model, used to simulate cascading failures in power grids. Separate runs of Cascades are performed for the different test power grids namely, IEEE24, IEEE39, UK, and IEEE118.

and therefore, becomes an instance of the dataset. For each combination of loading condition and element in the outage list, we simulate the cascading failure, identify the terminal state of the power grid, quantify the demand not served, and list the tripped elements. Figure 1 shows the structure of the Cascades model Gjorgiev & Sansavini (2022).

## 3  POWERGRAPH BENCHMARK FOR GRAPH-LEVEL PREDICTIONS AND EXPLAINABILITY

The PowerGraph dataset is obtained by processing the results of the Cascades model. Because we work with graph-level tasks, the dataset is a collection of $N$ attributed graphs $\mathcal{G} = \{G_1, G_2, .., G_N\}$. Each input graph reflects a unique pre-outage operating condition of the system and one set of single/multiple outages. Therefore, the total number of graphs $N$ per power grid equals to $n_{load\ cond} * n_{outage\ lists}$. Finally, each graph is assigned an output label corresponding to the chosen task. An attributed graph is defined $G = (\mathcal{V}, \mathcal{E}, \mathbf{V}, \mathbf{E})$, where $\mathcal{V}$ is the set of nodes (bus) and $\mathcal{E}$ is the set of edges (branches), $\mathbf{V} \in \mathbb{R}^{|\mathcal{V}| \times t}$ is the node feature matrix, with $|\mathcal{V}|$ nodes and $t$ features per node and $\mathbf{E} \in \mathbb{R}^{|\mathcal{E}| \times s}$ is the edge feature matrix, with $|\mathcal{E}|$ edges and $s$ features per edge. Finally, the graph connectivity information is encoded in COO format Fey & Lenssen (2019). We assign three bus-level features and four branch-level features. Each feature quantity is normalized using mean normalization. The input features are:

**Bus**:
- Net active power at bus i, $P_{i,net} = P_{i,gen} - P_{i,load}, P \in \mathbb{R}^{n_{bus} \times 1}$, where $P_{i,gen}$ and $P_{i,load}$ are the active generation and load, respectively.
- Net apparent power at bus i, $S_{i,net} = S_{i,gen} - S_{i,load}, S \in \mathbb{R}^{n_{bus} \times 1}$, where $S_{i,gen}$ and $S_{i,load}$ are the apparent generation and load, respectively.
- Voltage magnitude at bus i, $V_i \in \mathbb{R}^{n_{bus} \times 1}$, where $n_{bus}$ is the number of buses in the power grid.

**Branch**: Active power flow $P_{i,j}$, Reactive power flow $Q_{i,j}$, Line reactance $X_{i,j}$, Line rating $lr_{i,j}$.

Figure 2 displays an instance of the PowerGraph dataset. Each graph represents a state of the power grid associated with a loading condition and an outage (single or multiple failures). Since each outage is associated with disconnected branches, we remove the respective branches from the adjacency matrix and from their respective edge features. Therefore, each instance of the dataset is a graph with a different topology. The total number of instances is reported in Table 1. For each initial power grid state, we have knowledge of the post-outage evolution of the system, i.e., the demand not served (DNS) and the number of tripped lines. We label it as a cascading failure in each case that results in branches tripping after the initial outage. With these two results, we can assign an output label to each graph for different models:

Binary classification - we assign each instance to two classes:

Table 2: Multi-class classification of datasets. c.f. stands for *cascading failure* and describes a state resulting in cascading failure of components. DNS denotes demand not served.

| Category A | Category B | Category C | Category D |
|---|---|---|---|
| DNS > 0 MW | DNS > 0 MW | DNS = 0 MW | DNS = 0 MW |
| c.f. ✓ | c.f. × | c.f. ✓ | c.f. × |

Table 3: Results of categorization in percentage.

| Power grid | Category A | Category B | Category C | Category D |
|---|---|---|---|---|
| IEEE39 | 2.18% | 3.48% | 1.46% | 92.88% |
| IEEE118 | 0.07% | 5.84% | 2.01% | 92.08% |
| IEEE24 | 33.90% | 4.88% | 0.16% | 61.06% |
| UK | 4.06% | 0% | 8.02% | 87.92% |

- DNS=0, initial state results in a stable state, label 0
- DNS>0, initial state results in an unstable state, label 1

Multi-class classification - we assign each instance to four classes:

- DNS>0, cascading failure of components besides the first trigger, Category A
- DNS>0, no cascading failure of components besides the first trigger Category B
- DNS=0, cascading failure of components besides the first trigger, Category C
- DNS=0, no cascading failure of components besides the first trigger, Category D

Regression - we assign each instance the DNS in MW

The choice among binary classification, multi-class classification, or regression depends on the use of the GNN model trained with the PowerGraph dataset. The binary classification model serves as an early warning system, i.e., detects initial states of the power grid that are critical. The multi-class classification model allows us to distinguish different scenarios. Indeed, a transmission system operator could benefit from knowing when a cascading failure does not necessarily cause demand not served and vice-versa. Finally, with the regression model, we can directly access the final demand not served associated with particular pre-outage states of the system. In this case, the GNN model becomes a surrogate of the physics-based model useful both as an early warning system and to perform security evaluation with low computational cost.

**Explainability mask** We assign ground-truth explanations as follows: when a system state undergoes a cascading failure, the cascading edges are considered to be explanations for the observed demand not served. Therefore, for the Category A instances, we record the branches that fail during the development of the cascading event. We set the explainability mask as a Boolean vector $\mathbf{M} \in \mathbb{R}^{|\mathcal{E}| \times 1}$, whose elements are equal to 1 for the edges belonging to the cascading stage and 0, otherwise (see Figure 2).

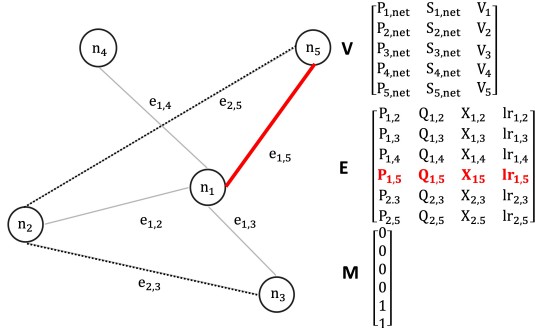

Figure 2: Structure of one instance of the GNN dataset for an exemplary power grid. The same structure is kept for all the power grids in PowerGraph, IEEE24, IEEE39, UK, and IEEE118. We highlight the initial outage in red, the line is removed both from the graph connectivity matrix and from the edge feature matrix. The cascading edges are highlighted with the dotted line and encoded in the $\mathbf{M}$ boolean vector (0 - the edge has not tripped during cascading development, 1 - otherwise).

## 4 BENCHMARKING GRAPH CLASSIFICATION AND REGRESSION MODELS

In this section, we outline the method used to benchmark classification and regression models.

**Experimental setting and evaluation metrics** For each power grid dataset, we utilize baseline GNN architectures as they are common in the graph xAI community. Specifically, we use GCN-Conv Kipf & Welling (2016), GATConv Veličković et al. (2018), and GINEConv Hu et al. (2020b) to demonstrate that the PowerGraph datasets can be used to benchmark GNN and methods used to explain them. Furthermore, we experimented with the state-of-the-art graph transformer convolutional layers Shi et al. (2020) since they are the backbones of the most recent Graph Transformer models: GraphGPS Rampášek et al. (2022), Transformer-M Luo et al. (2022), TokenGT Kim et al. (2022). Finally, we resort to all of the aforementioned models because they account for the edge features, which are highly relevant in the case of power grids. We tune the number of MPL $\in \{1, 2, 3\}$ and the hidden dimensionality $\in \{8, 16, 32\}$. Adam optimizer is used with the initial learning rate of $10^{-3}$. Each model is trained for 200 epochs with learning rate adjusted in the learning process using a scheduler, which automatically reduces the learning rate if a metric has stopped improving. We split train/validation/test with 80/10/10% for all datasets and choose a batch size of 128. We present three graph-level models, namely, binary/ multi-class classification, and regression. For classification models, we consider balanced accuracy Brodersen et al. (2010) as the reference evaluation metric. Indeed, balanced accuracy has been designed as a metric for classification tasks where a strong class imbalance is observed (see Table 3). It allows prioritizing all the classes equally, in contrast to the F1 or F2 score, and it gives interpretable results for multiclass classification, in contrast to ROC-AUC Saito & Rehmsmeier (2015). Indeed, a strong class imbalance is observed. For regression models, we use mean squared error as metric.

**Observations** We report the best model performance for each power grid and MPL in Tables 4, 5, and 6. For the different MPL, we only show the set of hyper-parameters yielding the best performance, and the best model per power grid is highlighted in bold. The GNN architecture comprises 1) a number of MPLs, each followed by PReLU He et al. (2015) activation function, 2) a global pooling operator to obtain graph-level embedding from node embeddings, and 3) one fully connected layer. For the classification model, we do not observe relevant differences among the mean, max, and sum global pooling operators. The classification results are obtained with max global pooling. The regression results are obtained by concatenating max and sum global poolings.

Table 4: Binary classification models results on the test set averaged over five random seeds. Balanced accuracy is used as reference metric.

| Power grid | MPL type | No MPL | Hidden dimension | Test Accuracy | Test Balanced Accuracy |
|---|---|---|---|---|---|
| **IEEE24** | GCN | 2 | 32 | $0.8667 \pm 0.0049$ | $0.8769 \pm 0.0056$ |
| | GINe | 3 | 32 | $0.9798 \pm 0.0046$ | $0.9800 \pm 0.0035$ |
| | GAT | 3 | 32 | $0.9008 \pm 0.0052$ | $0.9067 \pm 0.0034$ |
| | **Transformer** | **3** | **16** | $\mathbf{0.9907 \pm 0.0040}$ | $\mathbf{0.9910 \pm 0.0037}$ |
| **IEEE39** | GCN | 3 | 32 | $0.9733 \pm 0.0012$ | $0.8113 \pm 0.0011$ |
| | GINe | 2 | 32 | $0.9939 \pm 0.0020$ | $0.9550 \pm 0.0041$ |
| | GAT | 3 | 32 | $0.9697 \pm 0.0023$ | $0.7865 \pm 0.0061$ |
| | **Transformer** | **3** | **16** | $\mathbf{0.9952 \pm 0.0015}$ | $\mathbf{0.961 \pm 0.016}$ |
| **UK** | GCN | 3 | 32 | $0.9657 \pm 0.0027$ | $0.7176 \pm 0.0023$ |
| | **GINe** | **2** | **32** | $\mathbf{0.9975 \pm 0.0018}$ | $\mathbf{0.9820 \pm 0.0010}$ |
| | GAT | 3 | 8 | $0.9889 \pm 0.0005$ | $0.9175 \pm 0.0012$ |
| | **Transformer** | **3** | **16** | $\mathbf{0.9960 \pm 0.0016}$ | $\mathbf{0.9820 \pm 0.0045}$ |
| **IEEE118** | GCN | 3 | 32 | $0.9917 \pm 0.0015$ | $0.9364 \pm 0.0032$ |
| | GINe | 3 | 8 | $0.9992 \pm 0.0046$ | $0.9921 \pm 0.0035$ |
| | GAT | 3 | 32 | $0.9880 \pm 0.0012$ | $0.9427 \pm 0.0005$ |
| | **Transformer** | **3** | **32** | $\mathbf{0.9992 \pm 0.0005}$ | $\mathbf{0.9947 \pm 0.0041}$ |

**Discussion** Most GNN models achieve high performance on the power grids of PowerGraph. We compare GCN, GAT, GINe, and Transformer. Of all MPL considered, only GCN does not take edge features into account; as a result its performance is low in most cases. Transformer achieves the state-of-the-art on all power grids for the binary and multi-class models. In the regression model, Transformer and GINe are the best-performing models. Overall, the model for binary

Table 5: Multi-class classification models results on the test set averaged over five random seeds. Balanced accuracy is used as reference metric.

| Power grid | MPL type | No MPL | Hidden dimension | Test Accuracy | Test Balanced Accuracy |
|---|---|---|---|---|---|
| **IEEE24** | GCN | 2 | 32 | $0.8465 \pm 0.0023$ | $0.6846 \pm 0.0009$ |
| | GINe | 2 | 32 | $0.9798 \pm 0.0019$ | $0.9426 \pm 0.0028$ |
| | GAT | 3 | 32 | $0.9054 \pm 0.0020$ | $0.8375 \pm 0.0009$ |
| | **Transformer** | **3** | **32** | $\mathbf{0.9829 \pm 0.0012}$ | $\mathbf{0.9894 \pm 0.0016}$ |
| **IEEE39** | GCN | 2 | 8 | $0.9242 \pm 0.0019$ | $0.4071 \pm 0.0012$ |
| | GINe | 3 | 16 | $0.9939 \pm 0.0015$ | $0.9693 \pm 0.0019$ |
| | GAT | 2 | 16 | $0.9497 \pm 0.0022$ | $0.5577 \pm 0.0027$ |
| | **Transformer** | **3** | **32** | $\mathbf{0.9550 \pm 0.0009}$ | $\mathbf{0.9742 \pm 0.0016}$ |
| **UK** | GCN | 3 | 32 | $0.9068 \pm 0.0023$ | $0.4615 \pm 0.0038$ |
| | GINe | 2 | 32 | $0.9798 \pm 0.0020$ | $0.9347 \pm 0.0017$ |
| | GAT | 3 | 8 | $0.9563 \pm 0.0009$ | $0.7452 \pm 0.0014$ |
| | **Transformer** | **3** | **8** | $\mathbf{0.9912 \pm 0.0009}$ | $\mathbf{0.9798 \pm 0.0013}$ |
| **IEEE118** | GCN | 3 | 8 | $0.9771 \pm 0.0010$ | $0.8303 \pm 0.0016$ |
| | GINe | 3 | 32 | $0.9968 \pm 0.0018$ | $0.9586 \pm 0.0010$ |
| | GAT | 3 | 16 | $0.9677 \pm 0.0010$ | $0.7392 \pm 0.0011$ |
| | **Transformer** | **3** | **8** | $\mathbf{0.9992 \pm 0.0013}$ | $\mathbf{0.9833 \pm 0.0006}$ |

Table 6: Regression models results on the test set averaged over five random seeds. MSE error is used as reference metric.

| Power grid | MPL type | No MPL | Hidden dimension | MSE loss |
|---|---|---|---|---|
| **IEEE24** | GCN | 1 | 32 | $2.80\text{E-}03 \pm 5.69\text{E-}04$ |
| | GINe | 3 | 16 | $2.90\text{E-}03 \pm 2.88\text{E-}04$ |
| | GAT | 2 | 16 | $2.90\text{E-}01 \pm 5.00\text{E-}04$ |
| | **Transformer** | **3** | **8** | $\mathbf{2.70\text{E-}03 \pm 3.16\text{E-}04}$ |
| **IEEE39** | GCN | 2 | 32 | $5.61\text{E-}04 \pm 5.04\text{E-}05$ |
| | **GINe** | **3** | **32** | $\mathbf{5.04\text{E-}04 \pm 5.04\text{E-}05}$ |
| | GAT | 3 | 32 | $5.62\text{E-}04 \pm 4.66\text{E-}05$ |
| | Transformer | 3 | 32 | $5.47\text{E-}04 \pm 8.50\text{E-}05$ |
| **UK** | GCN | 3 | 32 | $7.07\text{E-}03 \pm 6.45\text{E-}04$ |
| | GINe | 2 | 32 | $7.65\text{E-}03 \pm 6.17\text{E-}04$ |
| | GAT | 3 | 32 | $7.60\text{E-}03 \pm 6.12\text{E-}04$ |
| | **Transformer** | **3** | **16** | $\mathbf{7.00\text{E-}03 \pm 5.10\text{E-}04}$ |
| **IEEE118** | GCN | 2 | 32 | $4.00\text{E-}06 \pm 2.94\text{E-}07$ |
| | **GINe** | **2** | **32** | $\mathbf{3.00\text{E-}06 \pm 3.51\text{E-}07}$ |
| | GAT | 2 | 8 | $4.00\text{E-}06 \pm 3.70\text{E-}07$ |
| | Transformer | 2 | 8 | $5.00\text{E-}06 \pm 6.55\text{E-}07$ |

and classification models exhibit excellent results. However, the regression model, which is of importance in providing a prediction of the demand not served, does not achieve the desired level of performance. While the classification models showed consistent performance across various power grids, the regression models demonstrate lower MSE values for larger power grids. This observation can be attributed to the fact that larger power grids offer a greater diversity of scenarios, thus making it increasingly more difficult for a GNN model to identify and learn cascading failure patterns. Nevertheless, a regression model offers the most informative and comprehensive results since it predicts the exact magnitude of demand not served given a component failure and operating conditions. However, our results show that the regression models trained on the PowerGraph datasets do not provide the expected performance. Therefore, further advancements and innovations in GNN architectures are needed to achieve more robust and accurate regression results. Finally, we test the capability of GNN model to generalize to the systems not seen in training, i.e. inductive property of GNN Vignac et al. (2020). We report the results in Appendix A.7.

Models trained using the above approach, although representing real systems, are built with synthetic data from a cascading failure model. To render these models applicable to real-world systems further work is necessary. First, the cascading failure model that generates the data needs to be validated

and calibrated on the system of interest. Second, the GNN model should be further trained using real-world cascading failure events from the system of interest.

## 5 BENCHMARKING EXPLANATIONS ON THE GRAPH-CLASSIFICATION MODELS

In this section, we outline the method used to benchmark explainability methods. We focus on explaining the power grids of Category A of the multi-class classification model. This choice is explained in Appendix A.2.

**Experimental setting and datasets** For each dataset, we take the trained Transformer with 3 layers and 32 hidden units described in section 4. To benchmark explainability methods, we do not necessarily need the best GNN model. An appropriate filtering on the nature of the predictions (correct or mix) and the focus of the explanation (phenomenon or model focus) Amara et al. (2022) can circumvent smaller test accuracy. We adopt the same training parameters. We evaluate the posthoc explainability methods: Saliency Baldassarre & Azizpour (2019), Integrated Gradient Sundararajan et al. (2017), Occlusion Faber et al., GradCAM Selvaraju et al. (2016), GNNExplainer Ying et al. (2019) with and without node feature mask, PGExplainer Luo et al. (2020), PGMExplainer Vu & Thai (2020), SubgraphX Yuan et al. (2021), and GraphCFE Ma et al. (2022). In Appendix A.3, we report more experimental details on the GNN performance and the explainability methods. The PowerGraph benchmark with explanations is used to test and compare existing explainability methods. The role of explainers is to identify the edges that are necessary for the graphs to be classified as Category A Amara et al. (2022). Then, the resulting edges are evaluated on how well they match the explanation masks, which represent the cascading edges. We compare the results obtained on the PowerGraph datasets with scores computed for the synthetic dataset BA-2Motifs Luo et al. (2020). See Appendix A.4 for more details. The comparison of PowerGraph to the BA-2Motifs dataset allows us to verify if our results align with state-of-the-art research on the explainability of GNN.

**Human-based evaluation** To evaluate the generated explanations, we use the balanced accuracy metric. It compares the generated edge mask to the ground-truth cascading edges and takes into account the class imbalance, i.e., cascading edges are a small fraction of the total edges. It measures how convincing the explanations are to humans. More details about this metric are given in Appendix A.5. We report the performance of 11 explainability methods on finding ground-truth explanations. All results are averaged on five random seeds. Accuracy scores are computed for the datasets in PowerGraph and the synthetic dataset BA-2Motifs.

**Model-centric evaluation** Human evaluation is not always practical because it requires ground truth explanations and can be very subjective, and therefore does not necessarily account for the model's reasoning. Model-focus evaluation however measures the consistency of model predictions w.r.t removing or keeping the explanatory graph entities. For more objective evaluation, we therefore evaluate the faithfulness of the explanations using the fidelity+ metric. The fidelity+ measures how necessary are the explanatory edges to the GNN predictions. For PowerGraph, edges with high fidelity+ are the ones necessary for the graph to belong to Category A. We compare the PowerGraph results with BA-2Motifs results, using the fidelity+ metric $fid_{+}^{acc}$. The $fid_{+}^{acc}$ is computed as in the GraphFramEx framework Amara et al. (2022) and described in Appendix A.6. We utilize GraphFramEx to compare explainability methods: we choose the *phenomenon* focus and the masks to be *soft* on the edges. Explanations are weighted explanatory subgraphs, where edges are given importance based on their contribution to the true prediction in the multi-class setting. Figure 4 reports the fidelity+ scores for the power grid datasets and for the synthetic dataset BA-2Motifs.

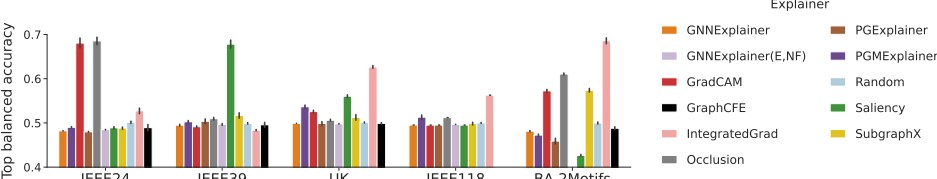

Figure 3: Top balanced accuracy of the PowerGraph datasets and the synthetic dataset BA-2Motifs. The *top* balanced accuracy is computed on explanatory edge masks that contain the *top* $k$ edges that contribute the most to the model predictions, with $k$ being the number of edges in the corresponding ground-truth explanations.

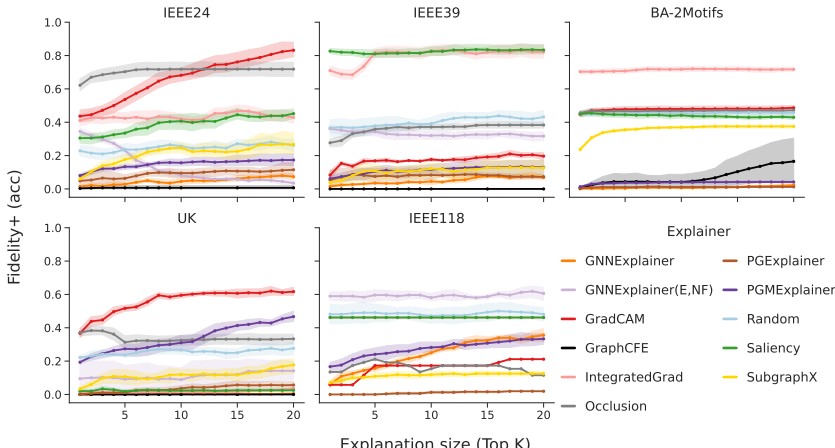

Figure 4: Faithfulness of the PowerGraph datasets and the BA-2Motifs dataset measured with the $fid+^{acc}$ metric as defined in Equation 2 in Appendix A.6. We conducted experiments on five random seeds. In the plot, alongside each data point, we have included confidence intervals calculated based on the standard deviation.

**Results**    Figure 3 shows that the best-balanced accuracies are obtained with the four methods, i.e., Saliency, Integrated Gradient, GradCAM, and Occlusion. Figure 4 also shows that these four methods have on average the highest fidelity+ on all datasets. Therefore, we conclude that they are the most appropriate methods to generate accurate and necessary explanations. Our observations on faithfulness are also consistent with previous results on the GraphFramEx benchmark Amara et al. (2022) that has already shown the superiority of gradient-based methods and Occlusion to return necessary explanations, i.e., the model predictions change when those explanatory entities are removed from the graph. However, in Figure 3 and Figure 4, no method globally outperforms the others for all datasets. For balanced accuracy, GradCAM and Occlusion are the best for IEEE24; Saliency for IEEE39; GradCAM for UK; and Integrated Gradient, Occlusion, GradCAM and SubgraphX for BA-2Motifs. On fidelity, GradCAM and Occlusion are the best for IEEE24; Saliency and Integrated Gradient for IEEE39; GradCAM for UK; and Integrated Gradient for BA-2Motifs. The choice of the optimal xAI method depends on the dataset. This is again consistent with the conclusions in Amara et al. (2022). Concerning the IEEE118 dataset, none of the methods is able to generate good explanations. The maximum top balanced accuracy is 0.55 and the maximum fidelity+ score is reached by GNNExplainer on edges and node features and is only 0.6. This performance is likely due to the complexity of the IEEE118. Being the largest power grid with 186 branches (see Table 1), the system contains complex interdependencies between the elements of the power grid during a cascading failure. As a consequence, node and edge-level features play a bigger role in explaining the GNN predictions. Therefore, we believe that an accurate model explanation will be obtained only with methods that provide node and link-level feature masks as well as edge masks. In addition, those methods could play a role in understanding the relevance of the input features to the GNN prediction, allowing to discard noisy features.

## 6    Conclusions

To strengthen the use of GNN in the field of power systems, we present PowerGraph, a dataset for graph-level tasks and model explainability. The dataset is suited to test graph classification and regression models. The main focus of PowerGraph is the analysis of cascading failures in power grids. Furthermore, experts often require interpretability of the results. Therefore, we benchmark the dataset for a variety of GNN and explainability models. The GNN models show excellent performance, in particular for graph classification, on our new benchmark, while graph regression models should be further developed. Finally, PowerGraph is the first real-world dataset with ground-truth explanations for graph-level tasks in the field of explainable AI. It allows us to evaluate both the accuracy and faithfulness of explainability methods in a real-world scenario. PowerGraph provides consistent outcomes that align with previous research findings and reinforce the concept that there is no universally superior method for explainability. In future work, we aim to extend the PowerGraph with new datasets  Birchfield et al. (2017) and include additional power grid analyses, including solutions to the power flow, the optimal power flow, and the unit commitment.

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
