# A    SUPPLEMENTARY MATERIALS

## A.1    OGB TAXONOMY OF GRAPH DATASETS

The Open Graph Benchmark Hu et al. (2020a) contains a diverse set of real-world datasets of various sizes and operational specifics. It contains medium to large-scale datasets that can be used to feed data-hungry models like GNN. For node and link property prediction tasks, OGB has datasets in all domains, *i.e.*, nature, society, and information. However, Table 7 shows the absence of graph datasets in the society domain. To fill this gap, we propose PowerGraph, the first collection of real datasets in the *society* domain.

Table 7: OGB taxonomy for graph datasets.

| Domain | Property prediction task | | |
|---|---|---|---|
| | **Node** | **Link** | **Graph** |
| **Nature** | `proteins` | `ddi`,`ppa` | `molhiv`,`molpcba/ppa` |
| **Society** | `arxiv`,`products`,`papers100M` | `biokg`,`wikikg2` | - |
| **Information** | `mag` | `collab`,`citation2` | `code2` |

## A.2    CLASS TARGETED EXPLANATIONS

For benchmarking explanations in section 5, we focus on explaining Category A graphs of the multi-class problem, i.e., the power grids that fail to serve the demand (DNS>0). The objective is to shed light on the lines that are tripped after the first contingency. We use the multi-class problem rather than the binary classification problem that classifies states according to the demand not served (DNS) only, i.e. distinguishes power grids that serve the demand (DNS=0, label 1) from the ones that do not (DNS>0, label 0). In the multi-class problem, the model learns to distinguish cascading failure scenarios, while in the binary setting, Category A and B are considered the same type of grids (class DNS>0). Choosing to explain DNS>0 in the multi-class problem allows us to focus on the case where some lines are tripped when DNS>0 and therefore expect the model to learn the cascading edges for this class of grids.

## A.3    EXPLAINABILITY METHODS

To explain the decisions made by the GNN models, we adopt different classes of explainers including gradient/feature-based methods and perturbation-based methods. In our experiments, we compare the following methods: **Random** gives every edge and node feature a random value between 0 and 1; **Saliency (SA)** measures node importance as the weight on every node after computing the gradient of the output with respect to node features Baldassarre & Azizpour (2019); **Integrated Gradient (IG)** avoids the saturation problem of the gradient-based method Saliency by accumulating gradients over the path from a baseline input (zero-vector) and the input at hand Sundararajan et al. (2017); **Grad-CAM** is a generalization of class activation maps (CAM) Selvaraju et al. (2016); **Occlusion** attributes the importance of an edge as the difference of the model initial prediction on the graph after removing this edge Faber et al.; **GNNExplainer (E,NF)** computes the importance of graph entities (node/edge/node feature) using the mutual information Ying et al. (2019); We also use **GNNExplainer** that considers only edge importance; **PGExplainer** is very similar to GNNExplainer, but generates explanations only for the graph structure (nodes/edges) using the re-parameterization mechanism to overcome computation intractability Luo et al. (2020); **PGM-Explainer** perturbs the input and uses probabilistic graphical models to find the dependencies between the nodes and the output Vu & Thai (2020); **SubgraphX** explores possible explanatory sub-graphs with Monte Carlo Tree Search and assigns them a score using the Shapley value Yuan et al. (2021); and **GraphCFE** leverages a graph variational autoencoder to generate counterfactual explanations for graphs Ma et al. (2022).

**Model-aware**. Gradient-based methods compute the gradients of target prediction with respect to input features by back-propagation. Features-based methods map the hidden features to the input space via interpolation to measure important scores. Decomposition methods measure the importance of input features by distributing the prediction scores to the input space in a back-propagation manner.

**Model-agnostic**. Perturbation-based methods use masking strategy in the input space to perturb the input. Surrogate models use node/edge dropping, BFS sampling and node feature perturbation. Counterfactual methods generate counterfactual explanations by searching for a close possible world using adversarial perturbation techniques Goodfellow et al. (2014).

Table 8: Explainability methods tested on the PowerGraph benchmark.

| Explainer | Model-aware/agnostic | Target | Type | Flow |
|---|---|---|---|---|
| SA | Model-aware | N/E | Gradient | Backward |
| IG | Model-aware | N/E | Gradient | Backward |
| Grad-CAM | Model-aware | N | Gradient | Backward |
| Occlusion | Model-agnostic | N/E | Perturbation | Forward |
| GNNExplainer | Model-agnostic | N/E/NF | Perturbation | Forward |
| PGExplainer | Model-agnostic | N/E | Perturbation | Forward |
| PGM-Explainer | Model-agnostic | N/E | Perturbation | Forward |
| SubgraphX | Model-agnostic | N/E | Perturbation | Forward |

## A.4 SYNTHETIC DATASET

In the explainability analysis of the paper, we compare PowerGraph datasets to the popular BA-2Motifs dataset. This dataset has 800 Barabási base graphs. Half graphs are attached with "house" motifs (label 0) and the rest are attached with five-node cycle motifs (label 1). The ground-truth explanations in this graph classification are the type of motifs attached to the base graph (house or five-node cycle). The BA-2Motifs dataset is commonly used to compare the performance of explainability methods Agarwal et al. (2023; 2022); Li et al. (2022); Longa et al. (2022); Yuan et al. (2022) because its ground truth explanations enable a simple interpretation for human-based evaluation.

## A.5 BALANCED ACCURACY

**Definition** The balanced accuracy is the arithmetic mean of the specificity and the sensitivity. The sensitivity or true positive rate or recall measures the proportion of real positives that are correctly predicted out of all positive predictions that could be made by the model. The specificity or true negative rate measures the proportion of correctly identified negatives over the total negative predictions that could be made by the model. The balanced accuracy is then expressed as:

$$\text{Balanced Accuracy} = \frac{\text{Sensitivity} + \text{Specificity}}{2} = \frac{1}{2} \cdot \left( \frac{TP}{TP + FN} + \frac{TN}{TN + FP} \right) \tag{1}$$

The balanced accuracy has the advantage of accounting for imbalance in the explanatory mask. In the context of cascading failure detection, we know that most of the components (links) in the grid will not fail. Therefore, the edge mask has many values that are zeros and only a few that are ones. The balanced accuracy measures if the method was able to recognize both failing and not failing edges, while giving the same importance to both detections.

## A.6 FAITHFULNESS METRIC

To measure the faithfulness of the explanations, we use either the fidelity- or the fidelity+ scores defined in Yuan et al. (2020); Amara et al. (2022). We evaluate the contribution of the produced explanatory subgraph to the initial prediction, either by giving only the subgraph as input to the model (fidelity-) or by removing it from the entire graph and re-run the model (fidelity+). As explained in section A.2, the generated explanations in the context of PowerGraph are the tripped lines and therefore should be necessary but not sufficient to the grid class. Indeed, the subgraph resulting from isolating the cascading branches does not represent a power grid. Therefore, fidelity- is not relevant in the context of the PowerGraph benchmark and we evaluate the faithfulness of explanations using the fidelity+ metric defined in equations 2 and 3. The fidelity score can be expressed either with probabilities ($fid_+^{prob}$) or indicator functions ($fid_+^{acc}$). We adopt the $fid_+^{acc}$, as it is more suitable for classification models. $f$ is a pre-trained classifier. We denote by $\hat{y}_i$ and $\hat{y}_i^{G_{C \setminus S}}$ the model's predictions when taking as input respectively the input graph $G_C$ and its complement or masked-out graph $G_{C \setminus S}$.

$$fid+^{acc} = \frac{1}{N} \sum_{i=1}^{N} | \mathbb{1}(\hat{y}_i = y_i) - \mathbb{1}(\hat{y}_i^{G_{C \setminus S}} = y_i) | \tag{2}$$

$$fid+^{prob} = \frac{1}{N} \sum_{i=1}^{N} \left( f(G_C)_{y_i} - f(G_{C \setminus S})_{y_i} \right) \tag{3}$$

## A.7 INDUCTIVE PROPERTY OF GNN MODELS ON POWERGRAPH

We conducted an out-of-distribution test by training GNN models on one power grid dataset and applying the model on a different power grid dataset. GNNs allow to train models that can be tested on grids with different topologies, as long as we feed the same number of features per node and edge. This attribute is often referred to as inductive learning property Vignac et al. (2020). We report the results in Tables 9, 10, 11. Table 9 shows that the binary classifier models trained on IEEE39, IEEE118, and UK datasets perform well on most datasets, except when tested on the IEEE24. Indeed, with a test balanced accuracy of 50%, these models are not able to identify patterns in IEEE24 and instead randomly assign instances to a class. Similarly, Table 10 indicates that the multiclass classification model trained on the IEEE39 achieves good performance across other power grid datasets, and in particular with the UK and IEEE118 datasets. However, Table 11 shows that the regression models yield identical MSE errors for all test sets. This behavior stems from the regression model assigning the same DNS value to all instances, indicating an inability to capture any structure in the test dataset. Overall, we conclude that the GNN models obtained from PowerGraph do not show robust results when applied on a different power grid dataset that the model did not observed during training.

Table 9: Out-of-distribution balanced accuracies of binary classification models. The selected model is the best performing model based on the Transformer MPL.

| Trained on \Tested on | IEEE24 Binary | IEEE39 Binary | UK Binary | IEEE118 Binary |
|---|---|---|---|---|
| IEEE24 Binary | 0.99 | 0.35 | 0.25 | 0.30 |
| IEEE39 Binary | 0.50 | 0.96 | 0.75 | 0.70 |
| UK Binary | 0.50 | 0.65 | 0.98 | 0.70 |
| IEEE118 Binary | 0.50 | 0.67 | 0.77 | 0.99 |

Table 10: Out-of-distribution balanced accuracies of multiclass classification models. The selected model is the best performing model based on the Transformer MPL.

| Trained on \Tested on | IEEE24 Multiclass | IEEE39 Multiclass | UK Multiclass | IEEE118 Multiclass |
|---|---|---|---|---|
| IEEE24 Multiclass | 0.98 | 0.071 | 0.12 | 0.0018 |
| IEEE39 Multiclass | 0.45 | 0.97 | 0.66 | 0.76 |
| UK Multiclass | 0.0072 | 0.048 | 0.98 | 0.067 |
| IEEE118 Multiclass | 0.0072 | 0.048 | 0.22 | 0.98 |

# B ACCESS TO POWERGRAPH DATASET

## B.1 DATASET DOCUMENTATION AND INTENDED USES

PowerGraph is the collection of the following GNN datasets: UK, IEEE24, IEEE39, IEEE118 power grids. We use `InMemoryDataset` Fey & Lenssen (2019) class of Pytorch Geometric, which processes the raw data obtained from the Cascades B. Gjorgiev (2019) simulation. For each dataset UK, IEEE24, IEEE39, IEEE118, we provide a folder containing the raw data organized in the following files:

- `blist.mat`: branch list also called edge order or edge index
- `of_bi.mat`: binary classification
- `of_reg.mat`: regression labels
- `of_mc.mat`: multi-class labels
- `Bf.mat`: node feature matrix
- `Ef.mat`: edge feature matrix
- `exp.mat`: ground-truth explanation

Table 11: Out-of-distribution MSE errors of regression models. The selected model is the best performing model based on the Transformer MPL.

| Trained on \Tested on | IEEE24 Regression | IEEE39 Regression | UK Regression | IEEE118 Regression |
|---|---|---|---|---|
| IEEE24 Regression | 2.70E-03 | 3.81E-04 | 3.81E-04 | 3.81E-04 |
| IEEE39 Regression | 1.73E-04 | 5.47E-04 | 1.73E-04 | 1.73E-04 |
| UK Regression | 9.89E-05 | 9.89E-05 | 2.34E-03 | 9.89E-05 |
| IEEE118 Regression | 9.44E-08 | 9.44E-08 | 9.44E-08 | 5.00E-06 |

## B.2 DOWNLOAD DATASET

The dataset can be viewed and downloaded by the reviewers from `https://figshare.com/articles/dataset/PowerGraph/22820534` (~1.8GB, when uncompressed):

```bash
#!/bin/bash
wget -O data.tar.gz "https://figshare.com/ndownloader/files/40571123"
tar -xf data.tar.gz
```

## B.3 AUTHOR STATEMENT

The authors state here that they bear all responsibility in case of violation of rights, etc., and confirm that this work is licensed under the CC BY 4.0 license.

## B.4 HOSTING, LICENSING, AND MAINTENANCE PLAN

The code to obtain the PowerGraph dataset in the `InMemoryDataset` Fey & Lenssen (2019) format and to benchmark GNN and explainability methods is available as a public GitHub repository at `https://anonymous.4open.science/r/PowerGraph/`. The authors are responsible for updating the code in case issues are raised and maintaining the datasets. We aim to extend the PowerGraph with new datasets and include additional power grid analyses, including solutions to the power flow, the optimal power flow, and the unit commitment. Over time we plan to release new versions of the datasets and provide updates to the results for both the GNN accuracy and the explainability analysis. In addition, the code will be updated if new pytorch/torch-geometric versions are released or crucial python packages are updated. The data is hosted on figshare at `https://figshare.com/articles/dataset/PowerGraph/22820534`. The authors give public free access to the PowerGraph dataset. The datasets are identified with the `DOI:10.6084/m9.figshare.22820534`. The work in this paper (code, data) is licensed under the CC BY 4.0 license.

## B.5 CODE IMPLEMENTATION

We run a hyper-parameters grid search over different GNN models, using torch-geometric 2.3.0 Fey & Lenssen (2019) and Torch 2.0.0 with CUDA version 11.8 Collobert et al. (2011); NVIDIA Corporation (2023). The experiments to benchmark graph classification and regression models are performed on a Windows machine with 3 GPUs NVIDIA RTX A6000 with 128 GB RAM memory. For the explainability analysis, experiments are conducted on 8 AMD EPYC 7742 CPUs with a memory of 5GB each on the ETH Euler clusters (CSCS).