# OpenReview forum: "PowerGraph: A power grid benchmark dataset for graph neural networks"
_ICLR.cc/2024/Conference — Submitted to ICLR 2024_

### Official Review · Reviewer_btRh · 2023-10-28

**Soundness:** 3 good
**Presentation:** 2 fair
**Contribution:** 2 fair
**Rating:** 5
**Confidence:** 3

**Summary:**

This paper proposes PowerGraph, a graph dataset that models cascading failures in power grids. PowerGraph is used for graph-level tasks for Graph Neural Networks(GNNs) in power grids domain. Power grid dataset is a good use case from the real-world, but PowerGraph is generated by domain models. The importance of the domain that PowerGraph covers is not discussed. Takeaway messages for experiments are unclear.

**Strengths:**

S1. Power grid graphs is a big real-world use case.

S2. Datasets are accessible to GNN community through PyG.

**Weaknesses:**

W1. It seems that the domain of PowerGraph,  providing datasets for graph-level tasks for GNN in power-grid, is limited. See Q1.

W2. It is confusing that PowerGraph is real dataset, as ground-truths are generated by physical based models. See Q2.

W3. Take away message of experiments is not clear. For example, the paper does not suggest which GNN models to use. See Q4.

W4. Paper is not easy to read.

**Questions:**

Q1. Table 7 in supplementary material shows that there are Nature, Society and Information domains for graph-level tasks. PowerGraph only covers society domain, which seems to be limited. Can authors give comments on this?

Q2: Can author explain that why PowerGraph is real dataset?

Q3: Physical based models are used to generated ground truths. What are the  limitations of  physical based models? How much should researchers and practitioners rely on generated ground truths?

Q4: Provide some insights for experiments.

Q5: Can authors give show computationally expense of offline simulations?

Q6: Paper says that existing models that are not based on ML are offline simulations and computationally expensive. Can GNN be a good
replacement technique?

Q7: What is the difference between PowerGraph and Texas A&M University Electric Grid Datasets https://electricgrids.engr.tamu.edu/?

Typos: Page 2, "GNN are convenient and" -> "GNNs are convenient and"

---

### Official Review · Reviewer_9FMu · 2023-10-29

**Soundness:** 2 fair
**Presentation:** 3 good
**Contribution:** 3 good
**Rating:** 3
**Confidence:** 3

**Summary:**

In this paper, authors build a new power grid dataset for graph neural networks. The dataset focuses on cascading failures and demand not served(DNS) and is built on the physics-based Cascades model. The dataset contains tasks of binary classification, multi-class classification and regression. The dataset is also assigned a ground-truth explanation so that it is possible to evaluate explanation methods on this new dataset. The comprehensive experiments are conducted to show that current graph neural networks can deal with binary classification and multi-class classification very well while it cannot handle the regression problem in this new dataset very well. Besides, the experiments of explanation methods show similar trends with previous dataset designed for explainability.

**Strengths:**

1. This is the first power grid benchmark dataset that focuses on cascading failures and DNS especially for GNNs.
2. The dataset contains a ground-truth explanation that can help for evaluating explanation methods.
3. The current unsatisfactory regression performance shows that this dataset can be used to guide improvement of future GNNs.

**Weaknesses:**

1. Only Category A data can have ground-truth explanations that constraint the usage of PowerGrid dataset to explanation methods.
2.  The dataset is built on a simulation model. And the gap between dataset and real-world scenarios are not properly demonstrated. Besides, authors mentioned that “the GNN model should be further trained using real-world cascading failure events from the system of interest.”, then how could this new dataset help in real-world scenarios?
3. The dataset seems similar to [1]. However, no comparison between the new dataset with [1] to show why the adjustment of construction is needed.

Reference:
[1] Varbella, Anna, Blazhe Gjorgiev, and Giovanni Sansavini. "Geometric deep learning for online prediction of cascading failures in power grids." Reliability Engineering & System Safety 237 (2023): 109341.

4. Some conclusions made in the paper seem wrong. For example,  authors demonstrate that “While the classification models showed consistent performance across various power grids, the regression models demonstrate lower MSE values for larger power grids. This observation can be attributed to the fact that larger power grids offer a greater diversity of scenarios, thus making it increasingly more difficult for a GNN model to identify and learn cascading failure patterns.” It seems that authors want to say larger power grids are more difficult to predict. However, a lower MSE(“the regression models demonstrate lower MSE values for larger power grids.”) indicates a better performance which means larger power grids are in fact easier to predict, which totally contradicts the author's statement.

**Questions:**

Besides the previous question in the weakness part, here are three additional questions for authors:
1. Why are the features of the buses introduced in the paper while the features of branches are not?
2. Is it possible to control the generation of the data so that each category can have balanced data numbers?
3. Other papers that aim at predicting the cascading failures generate their own dataset(Though they may not be designed for GNNs). Is it possible to compare PowerGraph with them?

Reference:

[1] Shuvro, Rezoan A., et al. "Predicting cascading failures in power grids using machine learning algorithms." 2019 North American Power Symposium (NAPS). IEEE, 2019.

[2] Zhou, Tianxin, Xiang Li, and Haibing Lu. "Power Grid Cascading Failure Prediction Based on Transformer." International Conference on Computational Data and Social Networks. Cham: Springer International Publishing, 2021.

---

### Official Review · Reviewer_RsGH · 2023-10-29

**Soundness:** 2 fair
**Presentation:** 3 good
**Contribution:** 3 good
**Rating:** 8
**Confidence:** 3

**Summary:**

This paper presents PowerGraph, a benchmark dataset tailored for GNNs in electrical power grid studies. GNNs have the potential to capture complex power grid phenomena and improve the accuracy of graph-level tasks such as identifying critical lines and predicting cascading failures. The lack of publicly available datasets for GNN applications in power grids has hindered progress in this area. PowerGraph aims to fill this gap by providing a large-scale, realistic dataset that can be used to train and evaluate GNN models for various tasks. The authors simulate a large number of cascading failure events on a synthetic power grid model. The resulting dataset contains over 10,000 graphs and can be used for both classification and regression tasks. Several baseline models are implemented for comparison and demonstrate the effectiveness of GNNs on PowerGraph. The dataset can also be used to benchmark GNN explainability methods as ground-truth edge-level explanations are also provided. Overall, the paper provides a valuable resource for researchers interested in applying GNNs to power grid analysis and highlights the potential of these models for improving the reliability and resilience of electrical power systems.

**Strengths:**

1. The proposed dataset benchmarks an important problem in the field of electrical power grids.

2. The authors provide a detailed description of the dataset generation process and propose several baseline models for comparison, which can serve as a useful starting point for future research.

3. The paper highlights the potential of GNNs for improving the accuracy of graph-level tasks in power grid analysis and demonstrates the effectiveness of these models on PowerGraph.

**Weaknesses:**

1. It is more like a suggestion than a weakness. Although the paper focuses on discussing GNNs, if I understand correctly, the dataset can be used to test models beyond the GNN framework (message-passing-based models) as well.

**Questions:**

N/A

---

### Official Review · Reviewer_vvXj · 2023-10-31

**Soundness:** 2 fair
**Presentation:** 3 good
**Contribution:** 2 fair
**Rating:** 3
**Confidence:** 4

**Summary:**

This paper introduces "PowerGraph," a GNN dataset for power grid analysis. It addresses the problem of lacking public datasets in this domain and enhances GNN applicability for analyzing power grid phenomena. By enabling GNNs to capture the complexity of power grids, it serves as a valuable resource for various fields and GNN model development.

**Strengths:**

1. The motivation is clear. The authors intend to tackle the issue of the absence of publicly available datasets in the field of power grids, thereby providing a valuable asset for research in the electrical power grid domain and graph-based machine learning.
2. The writing of the main context is well organized. It is easy for the reader to get to know the most critical intuition of the work.
3.  The experimental results are extensive, with the authors presenting outcomes from various perspectives, encompassing both classification and regression aspects.
4.  The code is released.

**Weaknesses:**

1. The dataset generation method is limited. The dataset generation process relies on physics-based cascading failure modeling. This approach may not capture all real-world scenarios accurately and could lead to discrepancies between simulated data and actual events. The dataset's quality and realism may be a limitation.
2. The application scenario lacks generality. The dataset and research focus on cascading failures in electrical power grids. However, there are no specific cascading failures discussed in this paper. This specificity may limit the broader applicability of the research. The insights and models developed may not easily transfer to other electrical power grid problems.
3. Lack of real data comparison. The work primarily relies on simulated data. A comparison with real-world data, where available, could strengthen the findings and insights.
4. While providing ground-truth edge-level explanations is a strength, it can also be a limitation. The process of assigning such explanations may introduce human bias and subjectivity. It is important to consider the accuracy of these explanations. However, the authors did not discuss it.

**Questions:**

1. What limitations are posed by the lack of public Graph Neural Networks (GNN) benchmark datasets for electrical power grids, and how do these datasets enhance the applicability of GNNs across different domains?
2. Could the author elaborate on how PowerGraph addresses the scarcity of historical blackout datasets and contributes to the understanding and research of electrical power grids, particularly in terms of cascading failure events and GNN training?
3. The author mentioned that PowerGraph has potential applications in various domains, such as chemistry and biology. In what ways do you see this dataset being applied in these other fields, and what advantages does it offer for these applications?

---

### Meta-Review · Area_Chair_6Rak · 2023-12-05

**Metareview:**

This paper introduces "PowerGraph," a novel dataset tailored for graph neural networks (GNNs) in the context of electrical power grid analysis. The dataset, focusing on cascading failures and demand not served (DNS), provides over 10,000 graphs for various tasks like classification and regression, alongside ground-truth explanations for better evaluation of GNNs. Although effective in certain classifications, the dataset reveals limitations in GNNs for regression tasks and the need for improved GNN explainability methods in the power grid domain.

While the proposed PowerGraph shows promising results, the reviewers have identified several weaknesses that need to be addressed:

1. The dataset's reliance on physics-based models for simulating cascading failures may not accurately reflect all real-world scenarios, potentially limiting its quality and realism.
2. The paper's specific focus on cascading failures in power grids, without discussing specific scenarios, may limit the broader applicability and transferability of its insights and models to other power grid problems.
3. There is no comparison with existing datasets, such as [1].

 [1] Varbella, Anna, Blazhe Gjorgiev, and Giovanni Sansavini. "Geometric deep learning for online prediction of cascading failures in power grids." Reliability Engineering & System Safety 237 (2023): 109341.

Based on these weaknesses, we recommend rejecting this paper. We hope this feedback helps the authors improve their paper.

**Justification For Why Not Higher Score:**

Most reviewers believed that the paper should be rejected, and furthermore, the authors did not provide any rebuttal comments during the author-reviewer discussion phase.

**Justification For Why Not Lower Score:**

N/A

---

### Decision · Program_Chairs · 2024-01-16

Reject